# A multi-organ point cloud registration algorithm for abdominal CT registration

Samuel Joutard[*,†], Thomas Pheiffer[†], Chloe Audigier[†],
Patrick Wohlfahrt[†], Reuben Dorent[*], Sebastien Piat[†],
Tom Vercauteren[*], Marc Modat[*], Tommaso Mansi[†]

[*] King's College London
[†] Siemens Healthineers

**Abstract.** Registering CT images of the chest is a crucial step for several tasks such as disease progression tracking or surgical planning. It is also a challenging step because of the heterogeneous content of the human abdomen which implies complex deformations. In this work, we focus on accurately registering a subset of organs of interest. We register organ surface point clouds, as may typically be extracted from an automatic segmentation pipeline, by expanding the Bayesian Coherent Point Drift algorithm (BCPD). We introduce MO-BCPD, a multi-organ version of the BCPD algorithm which explicitly models three important aspects of this task: organ individual elastic properties, inter-organ motion coherence and segmentation inaccuracy. This model also provides an interpolation framework to estimate the deformation of the entire volume. We demonstrate the efficiency of our method by registering different patients from the LITS challenge dataset. The target registration error on anatomical landmarks is almost twice as small for MO-BCPD compared to standard BCPD while imposing the same constraints on individual organs deformation.

## 1  Introduction

Registering CT images of the chest is an important step for several pipelines such as surgical planning for liver cancer resection or disease progression tracking [2, 1, 10, 15]. This step is both crucial and challenging as the deformations involved are large and may contain complex patterns such as sliding motion between organs. While traditional registration methods tend to fail on this task, learning approaches such as [12, 6, 5] obtained promising results at the Learn2Reg 2020 challenge, task 3 [7]. Yet, traditional and learning approaches both aims at registering the whole image content instead of focusing on the relevant structures of interests. This introduces undesired noise and complexity to the registration process. To tackle this issue, we propose to exploit the recent availability of high quality automatic segmentation pipelines such as [17, 3] and register the segmented structures. Specifically, structures are registered using their surface point cloud representation, allowing for exploiting meaningful geometric information of the different organs and finely modeling their dynamic properties. We

also stress that surface point clouds are easy to derive from segmentation masks and are a lightweight representation of the structures of interest.

The Coherent Point Drift [13] (CPD) algorithm is one of the most popular method for deformable point cloud registration considered as state of the art [11]. A recent work [9] extended this framework using a Bayesian formulation and obtained more robust performances. CDP and BCPD both assume that points move coherently as a group to preserve the structure coherence. This is mainly because these frameworks are designed to register point clouds representing a single object. Consequently, [13, 9] are not adapted for registering multi-organ points clouds. In particular, the coherency assumption doesn't stand for organs registration as each organ-specific point cloud may move independently to its neighbour, especially if we aim at registering inter-patient images.

In this work, we introduce a Multi-Organ Bayesian Coherent Point Drift algorithm (MO-BCPD) that models independent coherent structures. The contribution of this work is four-fold. Firstly, we extend the Bayesian formulation of CPD to model more complex structures interactions such as organ motion independence. Secondly, given that points clouds are obtained using automated segmentations, the proposed framework models partial segmentation errors allowing MO-BCPD to recover them. Thirdly, we model individual organ elasticity as part of the formulation. Fourthly, extensive experiments on 104 patients (10,712 pairs of patients) from the LiTS public dataset [4] demonstrate the effectiveness of our approach compared to BCPD. In particular, our method achieves an average target registration error on anatomical landmarks of 13mm compared to 22mm for the standard BCPD.

## 2   Method

In this section, we present our Multi-Organ Bayesian Coherent Point Drift algorithm. Let $\mathbf{y} = [\mathbf{y}_m]_{m \in \{1...M\}} \in \mathbb{R}^{M,3}$ be the source point cloud and $\mathbf{x} = [\mathbf{x}_n]_{n \in \{1...N\}} \in \mathbb{R}^{N,3}$ be the target point cloud where N and M are respectively the number of source and target points. We aim at finding the transformation $\mathcal{T}$ that realistically aligns these point clouds. In particular here, unlike in [13, 9], the considered point clouds both represent a set of organ surfaces. Hence, each point is associated with an organ. Let $\mathbf{l}^y = [l_m^y]_{m \in \{1...M\}} \in \{1 \ldots L\}^M$ be the organ labels of the source point cloud and $\mathbf{l}^x = [l_n^x]_{n \in \{1...N\}} \in \{1 \ldots L\}^N$ be the organ labels of the target point cloud. L is the number of organs.

*Transformation model.* Similarly to the BCPD, the Multi-Organ Bayesian Coherent Point Drift (MO-BCPD), decomposes the motion in two components: a similarity transform $\rho : \mathbf{p} \longrightarrow s\mathbf{Rp} + \mathbf{t}$ and a dense displacement field $\mathbf{v}$. Hence the deformed source point could is $[\mathcal{T}(\mathbf{y}_m)]_{m \in \{1...M\}} = [\rho(\mathbf{y}_m + \mathbf{v}_m)]_{m \in \{1...M\}}$. While this parametrization is redundant, [9] has shown that this makes the algorithm more robust to target rotations. Moreover, it is equivalent to performing a rigid alignment followed by a non-rigid refinement which corresponds to the common practice in medical image registration.

---

**Algorithm 1:** Multi-Organ BCPD $(\mathbf{y}, \mathbf{x}, \omega, \Lambda, B, S, U, \kappa, \gamma, \epsilon)$

---

$\mathbf{v} \leftarrow 0_{M,3},\ \Sigma \leftarrow Id_M,\ s \leftarrow 1,\ R \leftarrow Id_3,\ t \leftarrow 0_3,\ <\alpha_m> \leftarrow \frac{1}{M},$

$\sigma^2 \leftarrow \frac{\gamma}{D \sum_{m,n} u^y_{l^y_m,l^x_n}} \sum_{m,n} u^y_{l^y_m,l^x_n} \|x_n - y_m\|^2,\ \theta \leftarrow (\mathbf{v}, \alpha, \mathbf{c}, \mathbf{e}, \rho, \sigma^2),\ P \leftarrow \frac{1}{M} \mathbf{1}_{M,N}$

$\nu' \leftarrow 1_N,\ q_1(.,.) \leftarrow D^{\kappa \mathbf{1}_M} \phi^{0,\Sigma},$

$q_2(c,e) \leftarrow \prod_{n=1}^{N} (1-\nu'_n)^{1-c_n} \left(\nu'_n \prod_{m=1}^{M} \left(\frac{p_{mn}}{\nu'_n}\right)^{\delta_n(e_m)}\right)^{c_n},\ q_3(.,.) \leftarrow \delta_\rho \delta_{\sigma^2}$

**while** $L(q_1 q_2 q_3)$ *increases more than* $\epsilon$ **do**

> **Update $P$ and related terms:**
>
> $\forall m, n\ \phi_{m,n} \leftarrow u^y_{l^y_m,l^x_n} \phi^{y'_m, \sigma^2 Id_3}(x_n) \exp{-\frac{3s^2 \Sigma_{m,m}}{2\sigma^2}},$
>
> $\forall m, n\ p_{m,n} \leftarrow \frac{(1-\omega)<\alpha_m>\phi_{m,n}}{\omega p_{out}(x_n) + (1-\omega) \sum_{m'} <\alpha_{m'}>\phi_{m',n}},\ \nu \leftarrow P.1_N,\ \nu' \leftarrow P^T.1_M,$
>
> $\hat{N} \leftarrow \nu^T.1_M,\ \hat{\mathbf{x}} \leftarrow \Delta(\nu)^{-1}.P.\mathbf{x},$
>
> **Update displacement field and related terms:**
>
> $\Sigma \leftarrow \left(G^{-1} + \frac{s^2}{\sigma^2}\Delta(\nu)\right),\ \forall d \in \{1,2,3\}\ \mathbf{v}^d \leftarrow \frac{s^2}{\sigma^2}\Sigma\Delta(\nu)(\rho^{-1}(\hat{\mathbf{x}}^d) - \mathbf{y}^d),$
>
> $\mathbf{u} \leftarrow \mathbf{y} + \mathbf{v},\ <\alpha_m> \leftarrow exp\{\psi(\kappa + \nu_m) - \psi(\kappa M + \hat{N})\}$
>
> **Update $\rho$ and related terms:** $\bar{x} \leftarrow \frac{1}{\hat{N}} \sum_{m=1}^{M} \nu_m \hat{x}_m,\ \bar{\sigma}^2 \leftarrow \frac{1}{\hat{N}} \sum_{m=1}^{M} \nu_m \sigma_m^2,$
>
> $\bar{u} \leftarrow \frac{1}{\hat{N}} \sum_{m=1}^{M} \nu_m u_m,\ S_{xu} \leftarrow \frac{1}{\hat{N}} \sum_{m=1}^{M} (\hat{x}_m - \bar{x})(u_m - \bar{u})^T,$
>
> $S_{uu} \leftarrow \frac{1}{\hat{N}} \sum_{m=1}^{M} (u_m - \bar{u})(u_m - \bar{u})^T + \bar{\sigma}^2 Id_3,\ \Phi S'_{xu} \Psi^T \leftarrow svd(S_{xu}),$
>
> $R \leftarrow \Phi d(1,\ldots,1,|\Phi\Psi|)\Psi^T,\ s \leftarrow \frac{Tr(RS_{xu})}{Tr(S_{uu})},\ t \leftarrow \bar{x} - sR\bar{u},\ \mathbf{y'} \leftarrow \rho(\mathbf{y}+\mathbf{v})$
>
> $\sigma^2 \leftarrow \frac{1}{3\hat{N}} \sum_{d=1}^{3} \left((\mathbf{x}^d)^T \Delta(\nu')\mathbf{x}^d - 2\mathbf{x}^d P^T \mathbf{y'}^d + (\mathbf{y'}^d)^T \Delta(\nu)\mathbf{y'}^d\right) + s^2 \bar{\sigma}^2$
>
> **Update q:** $q_1(.,.) \leftarrow D^{\kappa \mathbf{1}_M} \phi^{\mathbf{v},\Sigma},$
>
> $q_2(c,e) \leftarrow \prod_{j=1}^{N} (1-\nu'_j)^{1-c_j} \left(\nu'_j \prod_{i=1}^{M} \left(\frac{p_{ij}}{\nu'_j}\right)^{\delta_i(e_j)}\right)^{c_j},\ q_3(.,.) \leftarrow \delta_\rho \delta_{\sigma^2}$

**end**

---

*Generative model.* As in [9], MO-BCPD assumes that all points from the target point cloud $[\mathbf{x}_n]_{n \in \{1\ldots N\}}$ are sampled independently from a generative model. A point $x_n$ from the target point cloud is either an outlier or an inlier which is indicated by a hidden binary variable $c_n$. We note the probability for a point to be an outlier $\omega$ (i.e. $\mathcal{P}(c_n = 0) = \omega$). If $x_n$ is an outlier, it is sampled from an outlier distribution of density $p_{out}$ (typically, a uniform distribution over a volume containing the target point cloud). If $x_n$ is an inlier ($c_n = 1$), $x_n$ is associated with a point $\mathcal{T}(y_m)$ in the deformed source point cloud. Let $e_n$ be a multinomial variable indicating the index of the point of the deformed source point cloud with which $x_n$ is associated (i.e. $e_n = m$ in our example). Let $\alpha_m$ be the probability of selecting the point $\mathcal{T}(y_m)$ to generate a point of the target point cloud (i.e. $\forall n\ \mathcal{P}(e_n = m | c_n = 1) = \alpha_m$). $x_n$ is then sampled from a Gaussian distribution with covariance-matrix $\sigma^2 Id_3$ ($Id_3$ is the identity matrix of $\mathbb{R}^3$) centered on $\mathcal{T}(\mathbf{y}_m)$. Finally, the organ label $l^x_n$ is sampled according to

the label transition distribution $\mathcal{P}(l_n^x|l_m^y) = u_{l_n^x, l_m^y}$. The addition of the label transition term is our contribution to the original generative model [9]. This term encourages to map corresponding organs between the different anatomies while allowing to recover from partial segmentation errors from the automatic segmentation tool.

We can now write the following conditional probability density:

$$p^e(x_n, l_n^x, c_n, e_n|\mathbf{y}, \mathbf{l}^y, \mathbf{v}, \alpha, \rho, \sigma^2)$$

$$= (\omega p_{out}(x_n))^{1-c_n} \left( (1-\omega) \prod_{m=1}^{M} \left( \alpha_m u_{l_m^y, l_n^x} \phi^{\mathbf{y}'_m, \sigma^2 Id_3}(\mathbf{x}_n) \right)^{\delta_{e_n=m}} \right)^{c_n} \quad (1)$$

where $\phi^{\mu, \Sigma}$ is the density of a multivariate Gaussian distribution $\mathcal{N}(\mu, \Sigma)$ and $\delta$ is the Kronecker symbol.

*Prior distributions* MO-BCPD also relies on prior distributions in order to regularize the registration process and obtain realistic solutions. As in [9], MO-BCPD defines two prior distributions: $p^v(\mathbf{v}|\mathbf{y}, \mathbf{l}_y)$ that regularizes the dense displacement field and $p^\alpha(\alpha)$ that regularizes the parameters $\alpha$ of the source point cloud selection multinomial distribution mentioned in the generative model. The prior on $\alpha$ follows a Dirichlet distribution of parameter $\kappa \mathbf{1}_M$. In practice, $\kappa$ is set to a very high value which forces $\alpha_m \approx 1/M$ for all $m$. To decouple motion characteristics within and between organs, we propose a novel formulation of the displacement field prior $p^v$. Specifically, we introduce 3 parameters: a symmetric matrix $S = [s_{l,l'}]_{l,l' \in \{0...L\}}$ and two vectors $\Lambda = [\Lambda_l]_{l \in \{0...L\}}$ and $B = [B_l]_{l \in \{0...L\}}$. The matrix $S$ parametrizes the motion coherence inter-organs. The vectors $\Lambda$ and $B$ respectively characterizes the variance of the deformation magnitude and motion coherence bandwidth within each organ. We define the displacement field prior for the MO-BCPD as:

$$p^v(\mathbf{v}|\mathbf{y}, \mathbf{l}^y) = \phi^{\mathbf{0}, G}(\mathbf{v}^1) \phi^{\mathbf{0}, G}(\mathbf{v}^2) \phi^{\mathbf{0}, G}(\mathbf{v}^3) \quad (2)$$

$$G = \left[ \Lambda_{l_i^y} \Lambda_{l_j^y} S_{l_i^y, l_j^y} \exp -\frac{\|y_i - y_j\|^2}{2B_{l_i^y} B_{l_j^y}} \right]_{i,j \le M} \quad (3)$$

Note that $G$ must be definite-positive, leading to strictly positive values for variance of the displacement magnitude $\Lambda_l$ and mild constraints on $S$.

*Learning.* Combining equations (1) and (2), the joint probability distribution of the variables $\mathbf{y}, \mathbf{l}^y, \mathbf{x}, \mathbf{l}^x, \theta$, where $\theta = (\mathbf{v}, \alpha, \mathbf{c}, \mathbf{e}, \rho, \sigma^2)$ is defined as:

$$p(\mathbf{x}, \mathbf{l}^x, \mathbf{y}, \mathbf{l}^y, \theta) \propto p^v(\mathbf{v}|\mathbf{y}, \mathbf{l}_y) p^\alpha(\alpha) \prod_{n=1}^{N} p^e(x_n, l_n^x, c_n, e_n|\mathbf{y}, \mathbf{l}_y, \mathbf{v}, \alpha, \rho, \sigma^2) \quad (4)$$

As in [9], we use variational inference to approximate the posterior distribution $p(\theta|\mathbf{x}, \mathbf{y})$ with a factorized distribution $q(\theta) = q_1(\mathbf{v}, \alpha) q_2(\mathbf{c}, \mathbf{e}) q_3(\rho, \sigma^2)$ so that

$q = \arg\min_{q_1, q_2, q_3} KL(q|p(.|\mathbf{x}, \mathbf{y}))$ where $KL$ is the Kullback–Leibler divergence. Similarly to [9], we derive the MO-BCPD algorithm presented in algorithm 1. The steps detailed in Algorithm 1 perform coordinate ascent on the evidence lower bound $L(\theta) = \int_\theta q(\theta) \ln \frac{p(\mathbf{x}, \mathbf{y}, \theta)}{q(\theta)} d\theta$. In algorithm 1, $\gamma$ is a hyper-parameter used to scale the initial estimation of $\sigma^2$ and $\epsilon$ is used for stopping criteria. We note $\Delta(\nu)$ the diagonal matrix with diagonal entries equal to $\nu$.

*Hyper-parameter setting.* The model has a large number of hyper-parameters which can impact the performance of the algorithm. Regarding $\kappa$, the parameter of the prior distribution $p^\alpha$, and $\gamma$, the scaling applied to the initial estimation of $\sigma^2$, we followed the guidelines in [9]. $\omega$ is set based on an estimate of the proportion of outliers on a representative testing set. Regarding $B$ and $\Lambda$, respectively the vector of organ-specific motion coherence bandwidth and expected deformation magnitude, they characterise organs elastic properties. Concretely, a larger motion coherence bandwidth $B_l$ increases the range of displacement correlation for organ $l$ (points that are further away are encouraged to move in the same direction). A larger expected deformation magnitude $\Lambda_l$ increases the probability of larger displacements for organ $l$. These are physical quantities expressed in $mm$ that could be set based on organs physical properties. The inter-organ motion coherence matrix $S$ should be a symmetric matrix containing values between 0 and 1. $S_{l,l} = 1$ for all organs $l \in \{1 \ldots L\}$ and $S_{l,l'}$ is closer to 0 if organs $l$ and $l'$ can move independently.

$u_{l,l'}$ is the probability that a point with label $l'$ generates a point with label $l$. As points labels are in practice obtained from an automatic segmentation tool, we note $[g_m^y]_{m \in \{1 \ldots M\}}$ and $[g_n^x]_{n \in \{1 \ldots n\}}$ respectively the unknown true organ labels of the source and target point clouds (as opposed to the estimated ones $[l_m^y]_{m \in \{1 \ldots M\}}$ and $[l_m^y]_{m \in \{1 \ldots M\}}$). We assume that points from the deformed source point cloud generate points with the same true labels (i.e. $\mathcal{P}(g_n^x = g_m^y | e_n = m) = 1$). Hence, the probability for $y_m$, with estimated organ label $l_m^y$ to generate a point with label $l_n^x$ is given by: $u_{l_n^x, l_m^y} = \sum_k p(g_m^y = k | l_m^y) p(l_n^x | g_n^x = k)$ where $p(g_m^y = k | l_m^y)$ is the probability that a point labelled $l_m^y$ by the automatic segmentation tool has true label $k$ and $p(l_n^x | g_n^x = k)$ is the probability that the automatic segmentation tool predicts the organ label $l_n^x$ for a point with true label $k$. These probabilities need to be estimated on a representative testing set. We note that if the segmentation is error-free, the formula above gives $U = Id_L$. Indeed, in that case the points organ labels correspond exactly to the organ true labels so a point belonging to a certain organ can only generate a point from the same organ in the target anatomy. Properly setting the organ label transition probability matrix $U$ is crucial to recover from potential partial segmentation errors. Fig. 1 illustrates with a toy example a situation where the algorithm converges to an undesired state if the segmentation error is not modeled properly.

*Interpolation.* Once the deformation on the organ point clouds is known, one might want to interpolate the deformation back to image space in order to resample the whole volume. As in [8] we propose to use Gaussian process regression to

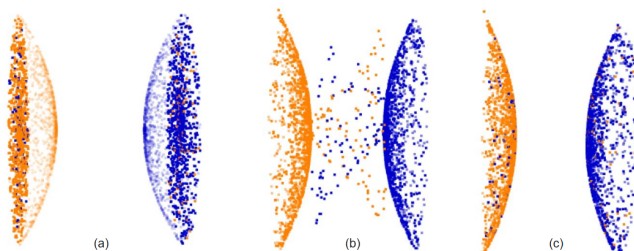

**Fig. 1.** Toy example registering a pair of organs (a blue and an orange organ) with ~10% segmentation error (corrupted input labels). Both organs (orange and blue) of the target point cloud are shown in (a) in transparent while the source point cloud is shown in opaque. The blue (orange) dots on the left (right) of the figure corresponds to simulated segmentation errors. (b) shows the registered point cloud without modeling the inter-organ segmentation error , (c) shows the registered point cloud with segmentation error modelization

interpolate the deformation obtained by the MO-BCPD algorithm. This interpolation process can also be used to register sub-sampled point clouds to decrease computation time as in [8].

Given a set of points $\tilde{\mathbf{y}} = [\tilde{y}_i]_{i \in \{1\ldots\tilde{M}\}}$ with labels $\mathbf{l}^{\tilde{y}} = [l_i^{\tilde{y}}]_{i \in \{1\ldots\tilde{M}\}}$. We compute the displacement for the set of points $\tilde{y}$ as:

$$\mathbf{v}^{\tilde{\mathbf{y}}} = G^{int}(\tilde{\mathbf{y}}, \mathbf{l}^{\tilde{y}}, \mathbf{y}, \mathbf{l}^y, B, \Lambda, S).G^{-1}.v \tag{5}$$

$$G^{int}(\tilde{\mathbf{y}}, \mathbf{l}^{\tilde{y}}, \mathbf{y}, \mathbf{l}^y, B, \Lambda, S)_{i,j} = \Lambda_{l_i^{\tilde{y}}} \Lambda_{l_i^y} S_{l_i^{\tilde{y}}, l_i^y} \exp -\frac{\|\tilde{y}_i - y_j\|^2}{2\beta_{l_i^{\tilde{y}}} \beta_{l_j^y}} \tag{6}$$

*Acceleration.* The speed ups mentioned in [9] section 4.6 are fully transferable to the MO-BCPD pipeline. In our experiments though, the main improvement, by far, came from performing a low rank decomposition of $G$ at the initialization of the algorithm. Indeed, this yielded consistent reliable $\times 10$ speed-ups with negligible error when using $\geq 20$ eigen values. The Nystrom methods to approximate $P$ sometimes implied large error due to the stochasticity of the method while yielding up to $\times 2$ speed-ups which is why we did not use it. This allows MO-BCPD to be run in a few seconds with $M, N \approx 5000$.

## 3   Experiments

We evaluate the MO-BCPD algorithm by performing inter-patient registration from the LITS challenge training dataset [4] which contains 131 chest-CT patient images. 27 patients were removed due to different field of view, issues with the segmentation or landmark detection. In total, 10,712 registrations were performed on all the pairs of remaining patients. The segmentation was automatically performed using an in-house tool derived from [17] which also provides

**Table 1.** Target registration error on landmarks. Results in mm (std).

|  | Sim | BCPD | GMC-MO-BCPD | OMC-MO-BCPD |
|---|---|---|---|---|
| bladder | 257 (26) | 29 (15) | 30 (15) | **26** (15) |
| left kidney bottom | 128 (18) | 23 (11) | 22 (10) | **8** (4) |
| left kidney center | 107 (13) | 18 (10) | 15 (8) | **6** (3) |
| left kidney top | 101 (15) | 23 (12) | 21 (10) | **9** (4) |
| liver bottom | 114 (15) | 29 (14) | 28 (14) | **24** (13) |
| liver center | 65 (10) | 12 (7) | 12 (7) | **11** (7) |
| liver top | 123 (15) | **24** (12) | 25 (12) | 26 (14) |
| right kidney bottom | 98 (15) | 26 (13) | 24 (11) | **10** (6) |
| right kidney center | 65 (11) | 21 (12) | 17 (9) | **5** (3) |
| right kidney top | 64 (15) | 25 (13) | 21 (11) | **9** (4) |
| round ligament of liver | 95 (20) | 27 (14) | 27 (13) | **25** (13) |

a set of anatomical landmarks for each image which were used for evaluation. We considered five organs of interest: the liver, the spleen, the left and right kidneys and the bladder. We compared 4 different algorithms: registration of the point clouds with a similarity transform (Sim), BCPD, GMC-MO-BCPD which is MO-BCPD with global motion coherence ($S = 1_{L,L}$) and OMC-MO-BCPD which is MO-BCPD with intra-organ motion coherence only ($S = Id_L$). As the segmentation tool performed very well on the considered organs, we set $U = Id_L$ and $\omega = 0$ for both MO-BCPD versions ($\omega = 0$ for BCPD as well). We used for all organs the same values for $\Lambda_l$ and $B_l$ respectively 10mm and 30mm as a trade off between shape matching and preservation of individual organs appearance ($\beta = 30$ and $\lambda = 0.1$ for BCPD which is the equivalent configuration). We also set, $\gamma = 1$ and $\epsilon = 0.1$. We compared those algorithms by computing the registration error on the anatomical landmarks belonging to those organs. We chose this generic, relatively simple setting (same rigidity values for all organs, no outlier modeling, only two extreme configurations for $S$) in order to perform large scale inter-patient registration experiments but we would like to stress that further fine tuning of these parameters for a specific application or even for a specific patient would further improve the modeling and hence the registration outcome. Results are presented in Table 1. We observe that while GMC-MO-BCPD induces some marginal improvements with respect to BCPD, OMC-MO-BCPD allows a much more precise registration. As illustrated in Fig. 2, the main improvement from BCPD to GMC-MO-BCPD is that different organs no longer overlap. Indeed, as highlighted by the green and blue ellipses, the spleen and the right kidney from the deformed source patient overlap the liver of the target patient when using BCPD. OMC-MO-BCPD properly aligns the different organs while preserving their shape (see orange and purple ellipses for instance).

## 4   Conclusion

We introduced MO-BCPD, an extension of the BCPD algorithm specifically adapted to abdominal organ registration. We identified three limitations of the

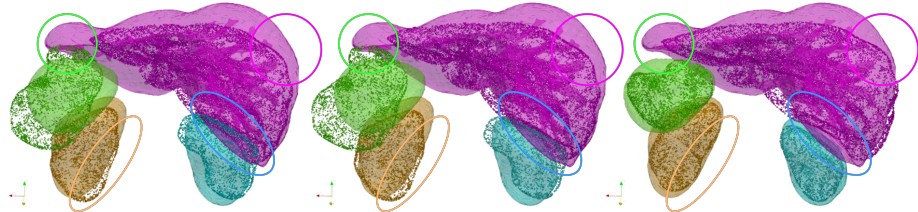

**Fig. 2.** Qualitative comparison of registration output for the liver, left/right kidneys and spleen. From left to right, BCPD, GMC-MO-BCPD, OMC-MO-BCPD. The target organs are shown in transparency while the deformed point cloud are shown in opaque.

original work [9] on this task and proposed solutions to model: the segmentation error between neighboring organs of interest, the heterogeneous elastic properties of the abdominal organs and the complex interaction between various organs in terms of motion coherence. We demonstrated significant improvements over BCPD on a large validation set (N=10,712).

Moreover, we would like to highlight that segmentation error could also be taken into account by tuning the outlier probability distribution $p_{out}$ and the probability of being an outlier $\omega$. When the point is estimated as a potential outlier by the algorithm, its contribution to the estimation of the transformation $\mathcal{T}$ is lowered. Hence, the segmentation error modelled by $p_{out}$ and $\omega$ corresponds to over/under segmentation, i.e. when there is a confusion between an organ and another class we don't make use of in MO-BCPD (e.g. background). Hence, MO-BCPD introduces a finer way of handling segmentation error by distinguishing two types of errors: mis-labeling between classes of interest which is modelled by $U$ and over/under-segmentation of classes of interest modeled by $\omega$ and $p_{out}$.

In this manuscript, we focused on highlighting the improvements yielded by the MO-BCPD formulation specifically designed for multi-organ point cloud registration. That being said, some clinical applications would require the deformation on the whole original image volume. Hence, we present in supplementary material preliminary results on a realistic clinical use case. In figures 3 and 4 we see that MO-BCPD coupled with the proposed interpolation framework obtain better alignment on the structures of interest than traditional intensity-based baselines. It is also interesting to note that MO-BCPD also better aligns structures that are particularly challenging to align such as the hepatic vein while not using these structures in the MO-BCPD.

Future work will investigate how MO-BCPD could be used as a fast, accurate initialization for image-based registration algorithms. From a modeling standpoint, we would also like to further work on segmentation error modeling (in particular over/under segmentation) with a more complex organ specific outlier distribution.

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

**Supplementary material:**
**A multi-organ point cloud registration algorithm**
**for abdominal CT registration**

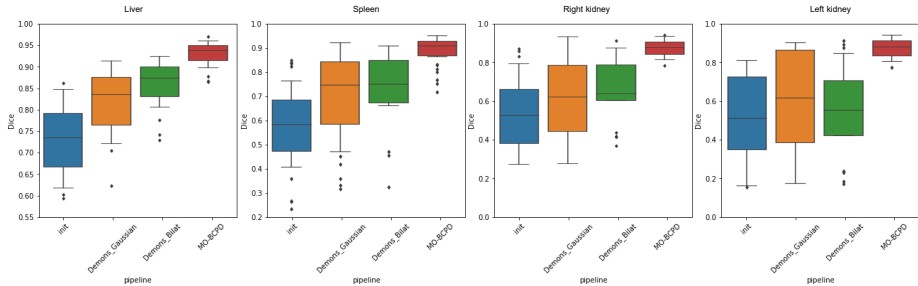

**Fig. 3.** One protocol for hepatocellular carcinoma patients diagnosis and follow up consist of acquiring CT scans of the patient liver at 4 different phases of diffusion of a contrast agent: before the diffusion, during the arterial phase, during the venous phase and  3minutes after the diffusion of the contrast agent. This process is repeated regularly to check the disease progression. This figure reports Dice scores on different structures when registering an image of a certain phase with the image of the same phase acquired during a follow-up scan of the same patient. In total, 31 registration were performed to compare 4 methods: no registration applied (init), the demons registration algorithm [16] (Demons_Gaussian), the demons registration algorithm with a bilateral filtering kernel [14] (Demons_bilat) and MO-BCPD using the proposed interpolation scheme to obtain the deformation on the whole image (MO_BCPD). We report results on 4 segmented structures used by MO-BCPD to give a general trend of the performances of the proposed method for whole volume registration. The 4 organs used by MO-BCPD are: the liver, the spleen and both kidneys. We see that, as expected, MO-BCPD coupled with the proposed interpolation framework better aligns those structures.

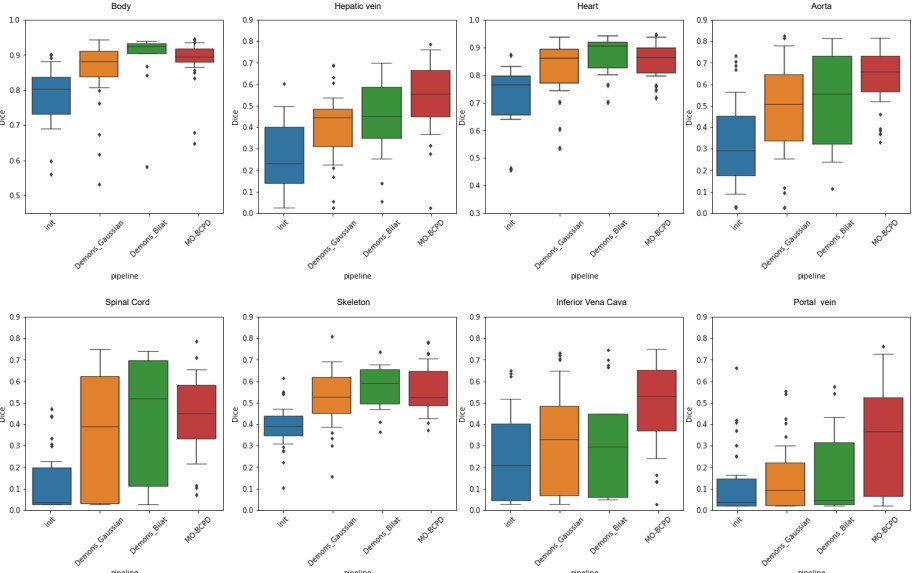

**Fig. 4.** In this figure, we report Dice scores on 8 structures not used by MO-BCPD from the same experiments as in figure 3: the body, the hepatic vein, the heart, the aorta, the spinal cord, the skeleton, the inferior vena cava and the portal vein. We see that, in general, MO-BCPD tends to achieve better or competitive results compared to the other baseline studied. In particular, we see that the interpolation of the MO-BCPD transformation to the whole volume gives good results for vessels alignments such as the hepatic vein or the portal vein compared to other baselines. This could be explain by the fact that those vessels motion is correlated with the one of the organs of interest used by MO-BCPD.