# OpenReview forum: "A multi-organ point cloud registration algorithm for abdominal CT registration"
_WBIR.info/2022/Workshop/Biomedical_Imaging_Registration — WBIR 2022_

### Official Review · Reviewer_Rsir · 2022-02-16

**Rating:** 4
**Confidence:** 3
**Recommendation:** Short Oral

**Deanonymize Review:**

no

**Detailed Comments:**

The paper presents a valuable multi-organ extension to the BCPD method, which allows to model independent coherent structures and individual organ elasticity. The authors conducted extensive experiments and demonstrate significant improvements over the underlying BCPD method.

Furthermore, the authors claim that the new method can model partial segmentation errors and allows to recover them. They demonstrate this on a toy experiment, but remove cases with segmentation issues from the experiment data and model the corresponding matrix as identity.

I recognize the valuable and novel multi-organ extension that the paper provides, but the paper is overloaded with technical details and I found it difficult to read and to understand the general idea and motivation of the method.

-	p. 5: It is a clear limitation that cases with segmentation issues are removed and $U$ is set to $U = Id_L$. Please discuss in paper.
-	p. 5: Another limitation is the generic parameter setting which was chosen to perform large scale experiments. The authors stress that further fine-tuning can improve results. It would be helpful to see this happen for one example.
-	The method is not compared to other methods other than the underlying one. If possible, please evaluate the method on the Learn2Reg data for task 3.
-	With so many different parameters, a list of all parameters would have been helpful.
-	p. 3: Algorithm 1 cannot be understood. Symbols are not explained and neither is the meaning. Also it is positioned two pages away from its description in the text.

-  p. 5: Typo in Learning (second to last sentence): Algorithm 1 with capital A
-  p. 5: Typo in Hyper-parameter setting (third to last sentence): a certain organ
-  p. 6: Typo/grammar in Acceleration (second to last sentence):
    - sometimes
    - why we did not use it   OR  why we did not make use of it


**Paper Type:**

methodological development

**Strengths Weaknesses:**


Strength:
-	good overview of previous work and how this work extends the existing BCPD method
-	figures and tables demonstrate advantages of the method
-	improvement of TRE in comparison to underlying method
-	promising preliminary results on a realistic clinical use case

Weaknesses:
-	paper is overloaded with technical details and describes a large number of parameters.
-	the authors claim that the method can model partial segmentations errors and allows to recover them. However, all cases with segmentation issues are explicitly removed from the experiment data so there is no evaluation of this feature on real data.
-	the toy example demonstrates the importance of properly modeling the segmentation error. The authors do not show how to estimate the probabilities on a testing set.
-	no comparison to other state of the art learning methods for image registration

---

### Official Review · Reviewer_kD2q · 2022-02-18

**Rating:** 4
**Confidence:** 4

**Deanonymize Review:**

no

**Detailed Comments:**

There are some minor issues:

(1) In the Section “Experiments”, the motion coherence bandwidth within each organ and the variance of the deformation magnitude were set to be constant values. Please clarify the reason that as constant fails to reflect the difference across organs.

(2) what does the registration of the point clouds with a similarity transform refer to? ICP or other method?

(3) What is the segmentation tool used? And how many points were derived from each organ for each subject?

(4) How to maintain the manifold of a source point cloud? Especially, when the target and source point cloud vary in the number of points, how to solve the distance for each point?

**Paper Type:**

both

**Strengths Weaknesses:**

Registering CT images is challenging because of the heterogeneous content of the human abdomen, implying complex deformations.
This work fully considers the heterogeneity across organs. Based on the original CPD algorithm, this work introduces the segmentation error to avoid the unnecessary overlap of the neighbor organs, and heterogenous motion coherence for each organ. The writing is easy to follow, and the experimental results validate the improvement as compared to the baseline CPD.

---

### Official Review · Reviewer_ccN5 · 2022-02-19

**Rating:** 4
**Confidence:** 3

**Deanonymize Review:**

no

**Detailed Comments:**


- It would be good to make very clear the differences between your method and the previous [9]. Many of the phrases referencing the work were regarding similarity, such as "As in [9]" and "Similarly to [9]". But in one sentence for instance, "The addition of the label transition term is our contribution to the original generative model [9]" – this is extremely crucial and should be highlighted.
- For the following sentence, "These probabilities need to be estimated on a representative testing set." – could more details be included?
- I believe additional information could be added about the LiTS challenge data used. For instance, information about the distributions of hospitals/institutions, etc would be useful.
- It was mentioned that an  in-house tool was used for segmentation [17] – a few additional sentences briefly describing this method would be helpful.
- In Table 1, is it possible to obtain sub millimeter accuracy? Please explain that the bolded results are the minimum value for each organ.
- How were the values β = 30 and λ = 0.1 chosen?
- In the methods it is mentioned that "But we would like to stress that further fine tuning of these parameters for a specific application or even for a specific patient would further improve the modeling and hence the registration outcome." – this should be expanded upon in the conclusion. How would one tune for a specific application or patient?
- Figure 2 is extremely crucial – but it was not explained enough. It would be beneficial to explain in detail what is shown in each ellipsis in each of the three figures. It would also be good to label your figure with sub captions for ease of understanding.
- Is your code publicly available? If not, it would be recommended to do so.

Overall, this was a very interesting manuscript that has exciting implications for future work. In general, additional details/more explanations would be helpful in the results section.


**Paper Type:**

methodological development

**Strengths Weaknesses:**

Strengths:
- The motivation and importance for the proposed algorithm is described sufficiently
- The methodology is explained in a very thorough and detailed manner

Weaknesses:
- The results section is a bit short and could be expanded. For instance, it might be interesting to visually see the deformation fields.
It would be beneficial to include some statistics to show that the difference between the target registration errors of your proposed method are significant compared to the other methods (Table 1). The same applies for the supplementary material.
- I am not certain that it is beneficial to include the results of registration using a similarity transform. It would be better to compare to coherent point drift [13] if possible
- I believe that the results, choices of parameters, etc require more detailed explanations.
- Future work can be lengthened.

---

### Official Review · Reviewer_Kk3r · 2022-02-21

**Rating:** 3
**Confidence:** 3

**Deanonymize Review:**

no

**Detailed Comments:**

The revisions needed:
- please provide more description to parameters & variables in the Algorithm 1.

- the results section - it would great to see some quantitative assessment of the segmentation accuracy (e.g. over segmentation, under segmentation) and its impact on the presented registration algorithm since modeling segmentation errors is one of the main novelties claimed in the paper. furthermore, it would be great to see also some results on the registration quality with respect of the organ size (segmentation of smaller and more complex shapes) such as hepatic veins.

 - the authors wrote in the introduction about sliding motion and chest imaging, while the paper is mostly focused on the abdominal organs (liver, kidneys), where the sliding motion is limited. It may be useful to test the presented algorithms also on publicly available data sets for lung registration (e.g. DirLab).

- the results illustrated in Figure 3. are not included in the Table 1. Would be possible to harmonise them (i.e. provide both quantitative results for the demons registration algorithm [16], the demons registration algorithm with a bilateral filtering kernel [14], and the proposed algorithm (and the baseline).

- what is the number of the extracted landmarks for each organ? Does the number of landmark impact on the registration performance? What is the minimum number for each organ to model independent motion?

- Page 8: significant improvement? provide p-value

Minor issues/comments:
 - Page 1: "(...) contain complex patterns such as sliding motion between organs. While traditional registration methods tend to fail on this task." - this is not clear what/why traditional methods fails on e.g. sliding motion? see [14] and reference therein for sliding motion estimation for CT images

- Labels in Figure 4 are too small (difficult to read)

eigen value -> eigenvalue
LiTS data set is NOT chest data set, it is abdomen/liver data set

**Paper Type:**

methodological development

**Strengths Weaknesses:**

The paper proposes an extension for Bayesian Coherent Point Drift to model motion (deformations) of multiple independent organs and to model the segmentation error. The registration is done by extracting points from the segmentations and then cloud of points is rigidly and non-rigidly registered.
the paper presents results on liver CT data set, and the new method's performance is favorable when compared to the baseline methods.
The paper is well-structured but some sections are difficult to read (see below), and some clarifications are needed before the paper can be accepted.

Strengths:
- an interesting and novel extension of Bayesian Coherent Point Drift (BCDP) to multi-organ BCPD (MO-BCPD) with a relevant application to medical imaging.

- The proposed MO-BCPD models independently different organ's deformations (motions) to capture different local motion patterns (e.g. sliding motion) using point clouds, and furthermore it models also errors coming from the organ's annotations (or automatic segmentations).

- the proposed algorithm is evaluated on liver CT and compared against the baseline method (BCPD), and its variants achieving the best overall results using the target registration error (TRE). the authors also shows some results comparing the intensity based dense registration against the proposed point-based registration.

Weaknesses:
- Section "Algorithm 1" is difficult to read and understand.  Please provide description of the input parameters and variables (what is what) in this section. it is difficult to analyse the algorithm if the parameters are not described in the algorithm section.

- missing assessment of segmentation performance and its impact on the registration performance. one of the claimed contributions claimed in this paper is to recover errors coming from the automatic segmentation algorithms, however such quantitative assessment is provided in the results section.

- the proposed algorithm has a number of hyper parameter that need to be optimized to achieve the-state-of-the-art performance. (although the authors have done nice job to describe their potential impact on the registration performance). Application to other organ (e.g. lungs) may help to understand how much these parameters impact the algorithm's performance.

---

### Decision · Program_Chairs · 2022-02-22

Accept